# AN EFFICIENT QUERY STRATEGY FOR ACTIVE LEARNING VIA OPTIMAL TRANSPORT

## ABSTRACT

Active Learning (AL) aims to reduce labeling costs by iteratively querying instances. Existing AL methods typically query instances based on either informativeness or representativeness. Only considering informativeness leads to sample bias. Only considering representativeness leads to query amount of instances before the optimal decision boundary is found. It is essential to consider both when querying instances. However, current hybrid methods are also time-consuming. To query instance efficiently while considering both informativeness and representativeness, we propose an efficient active query strategy based on optimal transport called Active Query by Optimal Transport (AQOT). Optimal Transport (OT) enables us to measure the difference between two distributions efficiently, allowing us considering the distribution of instances easily. Via entropy regularization, we can solve OT efficiently. Specifically, we make use of the sparseness of the solution of OT to querying the most informative instance while considering representativeness. Additionally, we introduce a dynamic adjustment to AQOT. By concatenating AQOT to multiple classification models, we show AQOT is a broadspectrum active query strategy. Experimental results demonstrate that our method surpasses state-of-the-art active learning methods and shows high efficiency.

## 1 INTRODUCTION

Many machine learning methods require a large number of labeled instances to train the model. Typically, these labels are assigned by human annotators, resulting in high labeling costs. In some domains of expertise like medical image recognition, data labeling is extremely expensive. Active learning is one of the main approaches to reduce the labeling cost (Settles, 2009). It continuously selects the most helpful instances to query from the oracles (e.g., human annotators) and aims to query as few instances as possible to improve the model most.

Due to the increasing demands of labeling data to train more complex models like deep neural network, active learning has received broad attention (Liu et al., 2022). Based on how we get unlabeled instances, active learning can be categorized into three scenarios (Settles, 2009). The first scenario is pool-based active learning, where all unlabeled instances are collected in a pool. We select query instances from the pool based on their utility (Lewis & Catlett, 1994). Pool-based active learning is a well-motivated scenario used in many machine learning tasks (Settles et al., 2008; Beluch et al., 2018). The second scenario is stream-based active learning (Zhu et al., 2007), where unlabeled instances are obtained from data stream. We must decide whether to query the instance once we get it. The last scenario is membership query synthesis, where query instances are generated based on the hypothesis model rather than being selected from existing unlabeled instances (Angluin, 1988; Tran et al., 2019). In this paper, we follow the pool-based active learning scenario.

When evaluating the utility of instances, most existing active learning strategies can be categorized into two main approaches: assessing informativeness and assessing representativeness. The former selects instances with the highest informativeness based on the assessment strategy, including entropy, distance and confidence (Guo & Schuurmans, 2007; Guo & Greiner, 2007; Bondu et al., 2010; Yang & Loog, 2016; Gal et al., 2017). Ning et al. (2022) proposed an active query method for open-set annotation based on uncertainty. Yan & Huang (2018) proposed an informative measurement for multi-labeling active learning, enhancing the adaptability of active learning methods.

Additionally, Yoo & Kweon (2019) proposed a target-agnostic loss prediction method to select samples that tasks are most uncertain. Furthermore, Konyushkova et al. (2017) introduced an approach centered on training a regressor that predicts the expected error reduction for candidate samples. Li & Guo (2013a) introduced a multi-label active learning strategy based on max-margin. However, the common issue of them is ignoring the distribution of all instances, which leads to sample bias when querying instances. The latter selects instances that represent the overall unlabeled instances. Two typical means to explore the representativeness are clustering methods and optimal experimental design methods (Brinker, 2003; Wu et al., 2006; Fu et al., 2013; Reitmaier et al., 2015; Ye & Guo, 2017). Wang & Ye (2015) proposed a batch-mode active learning strategy under empirical risk minimization principle, introducing techniques to select samples that enhance the overall representation. Sener & Savarese (2018) proposed a strategy to query diversity samples, particularly relevant in the context of convolutional networks. However, focusing on representativeness usually queries a lot of instances before we get close to the true decision boundary, which leads to large labeling cost.

It is essential to query instances taking both informativeness and representativeness into consideration. Many hybrid methods have been proposed (Huang et al., 2014; Li & Guo, 2013b). Sinha et al. (2019) proposed a hybrid method using a variational autoencoder and an adversarial network. Du et al. (2017) proposed a general active learning framework to fuse informativeness and representativeness. However, it is still time-consuming for current hybrid method to explore the representativeness of instances.

This paper proposes an efficient hybrid Active Query strategy by Optimal Transport called AQOT. Specifically, we establish two Optimal Transport (OT) models from unlabeled instances to positive instances and negative instances respectively. We design the active query strategy by examining the differences in the distributions of coefficient vectors between these two models. Furthermore, noticing that the quality of the solution is influenced by the initially labeled instances, we propose a dynamic adjustment to AQOT to encourage early exploration. AQOT outperforms state-of-the-art active learning methods with high efficiency. Besides, we empirically concatenate AQOT with mainstream classification models and verify it is a broad-spectrum strategy.

The rest of the paper is organized as follows. We introduce preliminaries in Section 2. Then we describe our approach in Section 3. Section 4 reports the experiments, followed by the conclusion in Section 5.

## 2 PRELIMINARY

Throughout the paper, we denote scalars by normal letters (e.g., y). We denote vectors and matrices by boldface lower and upper case letters respectively (e.g., $\boldsymbol{x}$ for vector and $\boldsymbol{X}$ for matrix). We denote by diag($\boldsymbol{a}$) the diagonal matrix with main diagonal equal to $\boldsymbol{a}$. We denote the $i$-th row and $j$-th column of $\boldsymbol{X}$ by $\boldsymbol{X}_{i:}$ and $\boldsymbol{X}_{:j}$ respectively. We denote sets by upper case letters with mathbb fonts (e.g., $\mathbb{X}$). For $\boldsymbol{X}, \boldsymbol{Y} \in \mathbb{R}^{m \times n}$, We denote by $\langle \boldsymbol{X}, \boldsymbol{Y} \rangle = \sum_{ij} X_{ij} Y_{ij}$. For positive integer $d$, we denote by $\mathbf{1}_d$ and $\Delta_d$ the $d$-dimensional all-one vector and $d$-dimensional simplex respectively. For positive integer $n$, we denote by $[n] = \{1, \ldots, n\}$.

### 2.1 OPTIMAL TRANSPORT

We transform an probability distribution into another distribution in OT problem. The goal is to minimize the total cost of the transform (Torres et al., 2021). By establishing OT model between two distributions, we can intuitively see their connections. We can easily take instance distribution into consideration when querying instances via OT.

We illustrate how OT works using a toy data set as an example. In this demonstration, we establish two OT models: one from the unlabeled instances to the positive instances and another from the unlabeled instances to the negative instances, just as we do in the following experiment. We treat each instance equally, i.e., assigning a probability value $1/u$ to each unlabeled instance, $1/p$ to each positive instance and $1/n$ to each negative instance. During the transporting process, unlabeled instances tend to transport to the nearest instance. The data set and coefficient matrix are shown in figure 1.

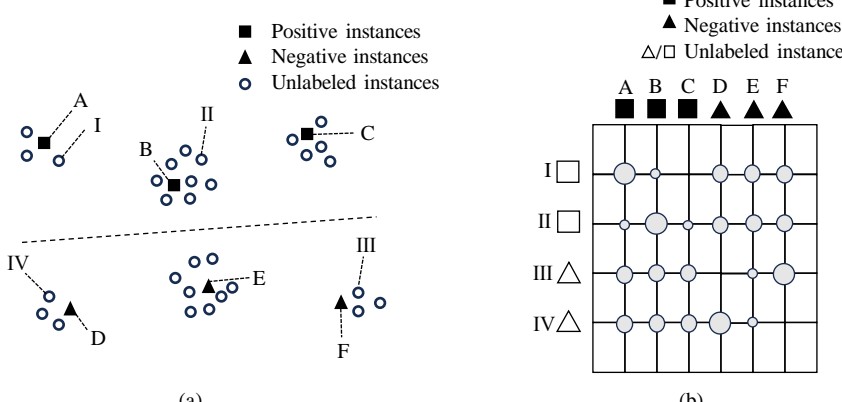

Figure 1: (a) The plot of a toy data set. Six labeled instances are marked from A to F and four example unlabeled instances are numbered from I to IV. The dashed line represents the decision boundary. (b) The coefficient matrix. The size of the circle represents the mass of unlabeled instances transported to labeled instances. I, II are positive instances indeed and III, IV are negative instances.

We denote by $\boldsymbol{u} \in \Delta_u$ and $\boldsymbol{l} \in \Delta_l$ two probability distributions respectively. The set of all admissible couplings $\mathcal{T}(\boldsymbol{u}, \boldsymbol{l})$ is:

$$\mathcal{T}(\boldsymbol{u}, \boldsymbol{l}) = \{\boldsymbol{T} \in \mathbb{R}_+^{u \times l} | \boldsymbol{T}\mathbf{1}_l = \boldsymbol{u}, \boldsymbol{T}^\top \mathbf{1}_u = \boldsymbol{l}\}, \tag{1}$$

where $\boldsymbol{T}$ is the coefficient matrix of this problem. $T_{ij}$ is the amount of mass transported from $u_i$ to $l_j$. We denote by $\boldsymbol{C} \in \mathbb{R}^{u \times l}$ the cost matrix. $C_{ij}$ is the cost of transporting unit mass from the position of $\boldsymbol{x}_i$ to the position of $\boldsymbol{x}_j$. We take Euclidean distance between two instances as the cost, which is $C_{ij} = \|\boldsymbol{x}_i - \boldsymbol{x}_j\|_2$. The goal of OT is to minimize the total transport cost from $\boldsymbol{u}$ to $\boldsymbol{l}$:

$$\min_{\boldsymbol{T} \in \mathcal{T}(\boldsymbol{u}, \boldsymbol{l})} \langle \boldsymbol{C}, \boldsymbol{T} \rangle = \min_{\boldsymbol{T} \in \mathcal{T}(\boldsymbol{u}, \boldsymbol{l})} \sum_{i \in [u]} \sum_{j \in [l]} C_{ij} T_{ij}. \tag{2}$$

Though equation (2) can be solved by any linear programming algorithm (Kantorovitch, 1958), the computational cost to precisely solve it in large scale problem is unacceptable. To address this, entropy regularization has been introduced (Cuturi, 2013), enabling faster and satisfying solutions to the entropy regularized OT problem:

$$\mathrm{OT}(\boldsymbol{u}, \boldsymbol{l}) = \min_{\boldsymbol{T} \in \mathcal{T}(\boldsymbol{u}, \boldsymbol{l})} \langle \boldsymbol{C}, \boldsymbol{T} \rangle - \lambda \cdot H(\boldsymbol{T}), \tag{3}$$

where $H(\boldsymbol{T}) = -\sum_{ij} T_{ij}(\log T_{ij} - 1)$ and $\lambda$ is regularization parameter. A larger $\lambda$ encourages a uniform coefficient distribution.

## 3 APPROACH

In this section, we describe the approach. Specifically, we first solve the entropy regularized OT problem by Sinkhorn-Knopp algorithm (Cuturi, 2013). We use the standard deviation of coefficient vectors to reflect the certainty in unlabeled instances. Then we propose our active query strategy via it. Finally, we propose the improvement of the active query strategy with dynamic adjustment.

We denote by $\mathbb{D}$ the data set with $n$ examples, which includes a labeled set $\mathbb{L} = \{(x_1, y_1), (x_2, y_2), \cdots, (x_{n_l}, y_{n_l})\}$ with $n_l$ labeled instances and an unlabeled set $\mathbb{U} = \{x_{n_l+1}, x_{n_l+2}, \cdots, x_{n_l+n_u}\}$ with $n_u$ unlabeled instances, where $n = n_l + n_u$. Besides, $y_i \in \{0, 1\} =: \mathbb{Y}$ is the ground-truth label and $x_i \in \mathbb{R}^d$ ($i \in [n]$). $\mathbb{L} = \mathbb{P} \cup \mathbb{N}$, where $\mathbb{P}$ and $\mathbb{N}$ denote the positive set with $n_p$ positive instances and negative set with $n_n$ negative instances respectively.

Active learning iteratively selects the most useful instance from $\mathbb{U}$ to query its label from the oracle. According the ground-truth label of the query instances, we add it to $\mathbb{P}$ or $\mathbb{N}$. Then we train the

classifier $F_\theta(\boldsymbol{x}) : \mathbb{R}^d \to \mathbb{Y}$ parameterized by $\theta$ with the updated $\mathbb{L}$. The classifier is expected to achieve better performance with the update of $\mathbb{L}$.

## 3.1 SOLVE OT

As mentioned before, we establish two OT models from $\mathbb{U}$ to $\mathbb{P}$ and $\mathbb{N}$ respectively in our experiment. We introduce entropy regularization into original OT problem and use Sinkhorn-Knopp algorithm to solve the entropy regularization OT problem. The Lagrangian of equation (3) with dual variables $\boldsymbol{\gamma} \in \mathbb{R}^{n_u}, \boldsymbol{\zeta} \in \mathbb{R}^{n_l}$ is:

$$L(\boldsymbol{T}, \boldsymbol{\gamma}, \boldsymbol{\zeta}) = \sum_{j \in [n_u]} \sum_{i \in [n_l]} (T_{ij} C_{ij} + \lambda T_{ij} (\log T_{ij} - 1)) + \boldsymbol{\gamma}^\top (\boldsymbol{T} \mathbf{1}_{n_l} - \frac{\mathbf{1}_{n_u}}{n_u}) + \boldsymbol{\zeta}^\top (\boldsymbol{T}^\top \mathbf{1}_{n_u} - \frac{\mathbf{1}_{n_l}}{n_l}).$$
(4)

By setting the partial derivative to zero, we can get the solution $\boldsymbol{T} = \mathrm{diag}(\boldsymbol{a}) \boldsymbol{K} \mathrm{diag}(\boldsymbol{b})$, where $\boldsymbol{a} = \exp(\boldsymbol{\gamma}/\lambda), \boldsymbol{K} = \exp(-\boldsymbol{C}/\lambda)$ and $\boldsymbol{b} = \exp(\boldsymbol{\zeta}/\lambda)$ are the element-wise exponential of $\boldsymbol{\gamma}/\lambda, -\boldsymbol{C}/\lambda, \boldsymbol{\zeta}/\lambda$. Considering the row and column marginals of $\boldsymbol{T}$ are equals to their target values, we have:

$$\boldsymbol{a} \odot (\boldsymbol{K}\boldsymbol{b}) = n_u, \boldsymbol{b} \odot (\boldsymbol{K}^\top \boldsymbol{a}) = n_l,$$

where $\odot$ is Hadamard product.

The heat maps of the OT coefficient matrix of stock are shown in figure 2. It is evident that the coefficient vector is sparse when transporting unlabeled instance to instances with same label, while more uniform when transporting to instances with different label. Based on this observation, we can use the coefficient matrix to assess the informativeness of an unlabeled instance and incorporate it into our active query strategy.

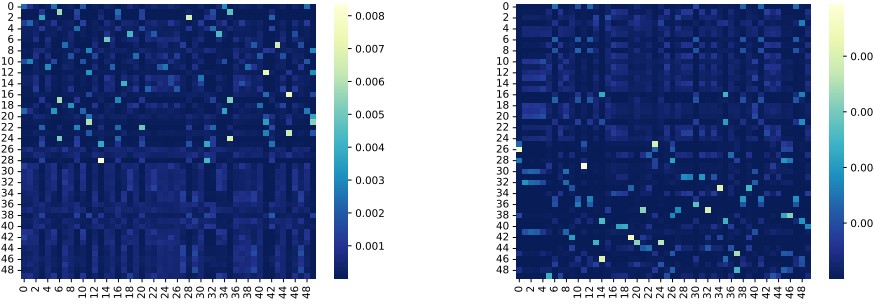

Figure 2: The heat maps of the OT coefficient matrix of stock. The OT models are established from 50 unlabeled instances to 50 positive instances and 50 negatives instances respectively from stock. Each row represents the coefficient vector of an unlabeled instance. The first half are positive instances and the second half are negative instances. It is evident that the coefficient vector is sparse when transporting to instances with same label.

## 3.2 QUERY STRATEGY

After establishing and solving the entropy regularization OT model, we obtain coefficient matrices $\boldsymbol{T}^P \in \mathbb{R}^{n_u \times n_p}, \boldsymbol{T}^N \in \mathbb{R}^{n_u \times n_n}$ for transporting unlabeled instances to positive and negative instances respectively. We can select the query instance with these two matrices. The dynamic query score for selecting query instance is:

$$\mathrm{dyscore}(\boldsymbol{x}_i) = (1 - \eta) \cdot (\alpha \cdot \mathrm{conf}_i + (1 - \alpha) \cdot \frac{1}{p_i}) + \eta \cdot d_i,$$
(5)

which consists of three terms: $\mathrm{conf}_i$, $1/p_i$ and $d_i$. We detail them respectively.

The first term represents our confidence in $\boldsymbol{x}_i$. For an unlabeled instance $\boldsymbol{x}_i \in \mathbb{U}$, its coefficient vectors are $\boldsymbol{T}_{i:}^P$ and $\boldsymbol{T}_{i:}^N$ respectively. $T_{ij}^P$ represents the mass transporting from $\boldsymbol{x}_i$ to $\boldsymbol{x}_j \in \mathbb{P}$ and $T_{ik}^N$ represents the mass transporting from $\boldsymbol{x}_i$ to $\boldsymbol{x}_k \in \mathbb{N}$. As previously introduced, the sparseness degree of the coefficient vector is related to the the label of source instance and target instance. The sparser the coefficient vector is, the more likely the source instance and the target instance share the same label. Standard deviation is a good method to reflect the the sparseness degree of instances. We define sample confidence of $\boldsymbol{x}_i$ by the standard deviation of coefficient vector:

$$\text{conf}_i = \left| \frac{\text{std}(\boldsymbol{T}_{i:}^P)}{\max(\boldsymbol{T}_{i:}^P)} - \frac{\text{std}(\boldsymbol{T}_{i:}^N)}{\max(\boldsymbol{T}_{i:}^N)} \right|. \tag{6}$$

Normalizing the standard deviation helps eliminate the influence of specific coefficients. When sample confidence is high, it is more likely that one coefficient vector is sparse while the other is uniform and we are more certain about the label of the instance. High confidence indicates certainty, which is a significant aspect of informativeness. Importantly, with the introduction of OT, this certainty takes the entire distribution of labeled instances into account. However, querying instances based solely on certainty is not sufficient. In addition to certainty, uncertainty is also a crucial factor in assessing the informativeness of instances. That is why we introduce the second term.

The second term represents the uncertainty degree of $\boldsymbol{x}_i$. In binary classification problems, $p_i = |p(y=1|\boldsymbol{x}_i) - p(y=0|\boldsymbol{x}_i)|$. In multi-class classification problems, $p_i = |p(y=\hat{y_1}|\boldsymbol{x}_i) - p(y=\hat{y_2}|\boldsymbol{x}_i)|$, where $\hat{y_1}$ and $\hat{y_2}$ represents the two labels with highest posterior probability. $p(y|\boldsymbol{x})$ can be computed with function predict_proba() in sklearn. Obviously, the smaller $p_i$ is, the more uncertain we are about the label of instance, making $p_i$ a good method to measure uncertainty. Ideally, we aim to query instances with both high uncertainty and high certainty. And we define query score based on uncertainty and certainty of the instance:

$$\text{score}(\boldsymbol{x}_i) = \alpha \cdot \text{conf}_i + (1-\alpha) \cdot \frac{1}{p_i}, \tag{7}$$

where $\alpha$ is the parameter to weigh uncertainty and certainty. In initial experiment, we simply set $\alpha = 0.5$, where certainty and uncertainty carry equal weight. Instances with high query scores have both high certainty and uncertainty, which is helpful to improve our classifier.

Based on the specific setup, active learning begins with a few labeled instances. If initial labeled instances are far away from the true decision boundary, it might take many iterations to get reach to the boundary. In some cases, querying instances near the wrong boundary reinforces the incorrect decision boundary. Encouraging exploration beyond the labeled instances leads to a quicker adjustment of the decision boundary, which is beneficial for achieving a potentially better boundary. So we propose a dynamic adjustment $d_i = \sum_{\boldsymbol{x} \in \mathbb{L}} |\boldsymbol{x}_i - \boldsymbol{x}|$ to the initial query score.

After adding the dynamic adjustment term, we get equation (5), where $\eta = 1/(\delta + \log(t))$, $t$ is current iteration, $\delta$ is the smooth parameter. At the beginning of training, $\eta$ is set close to 1, which encourages querying instances that are far from the labeled instances. This leads to rapid changes in the decision boundary, which might potentially get closer to the ground truth. In the worst case, it might result in a waste of the first few turns. However, $\eta$ decreases as training progresses and the query score of $\boldsymbol{x}_i$ is favored for querying instances. We prefer to querying instances with both certainty and uncertainty rather than outliers. In conclusion, in active query strategy with dynamic adjustment, we will query instance with the highest dynamic query score:

$$\boldsymbol{x}^{\text{query}} = \max_{\boldsymbol{x} \in \mathbb{U}} \text{dyscore}(\boldsymbol{x}). \tag{8}$$

The first term controls certainty. The second term controls uncertainty. The last term encourages exploration in the beginning of the training process. It is important to note that the computation of the score is independent of the specific classifier, allowing us to concatenate the query score with any classifiers. The algorithm is detailed in algorithm 1.

## 4 EXPERIMENTS

In this section, we concatenate AQOT strategy with three classification classifiers, i.e., SVM, GBDT and NN. We will begin by describing 6 real-world data sets, 6 compared methods, and experimental

---

**Algorithm 1** AQOT

---

**Input**: Initial $\mathbb{U}, \mathbb{P}, \mathbb{N}$, max query iteration T, $\delta$
**Output**: $F_\theta(\boldsymbol{x})$

1: $t \leftarrow 1$
2: $\eta \leftarrow 1/\delta$
3: **while** $t < T$ **do**
4:     Train a new classifier by $\mathbb{P}$ and $\mathbb{N}$.
5:     Establish OT models from $\mathbb{U}$ to $\mathbb{P}$ and $\mathbb{N}$ and compute the solution of the entropy-regularized OT problem by Sinkhorn-Knopp algorithm.
6:     **for** $i \leftarrow 1$ to $n_u$ **do**
7:         $\boldsymbol{T}_{i:}^P \leftarrow$ the coefficients vector of $\boldsymbol{x}_i \in \mathbb{U}$ transporting to $\mathbb{P}$
8:         $\boldsymbol{T}_{i:}^N \leftarrow$ the coefficients vector of $\boldsymbol{x}_i \in \mathbb{U}$ transporting to $\mathbb{N}$
9:         $p_i \leftarrow |p(y=1|\boldsymbol{x}_i) - p(y=0|\boldsymbol{x}_i)|$
10:     **end for**
11:     Query $\boldsymbol{x}^{\text{query}}$ according to equation (8).
12:     Add $\boldsymbol{x}^{query}$ to $\mathbb{P}$ or $\mathbb{N}$ according to its label and remove it from $\mathbb{U}$.
13:     $t \leftarrow t + 1$
14:     $\eta \leftarrow 1/(\delta + \log(t))$
15: **end while**

---

settings. Subsequently, we will assess the algorithm is not sensitive to entropy regularization parameter. In addition to comparing with the state-of-the-art active learning methods, we compare run time and AQOT shows high efficiency compared to other hybrid methods. Moreover, we conduct ablation experiments to show the effectiveness of AQOT.

## 4.1 EXPERIMENT SETTING

We utilize six data sets from the UCI Machine Learning Repository, including monks-problem-1, qsar-biodeg, balance-scale, phoneme, stock and breast. We compare the following query strategies in our work:

- AQOT: The proposed method of this paper, which queries the instance with high certainty and uncertainty as well as encourages exploring at the start of training.

- FULL: We train a classifier using all labeled instances as a reference baseline.

- RANDOM: This method queries instance randomly.

- UNCERTAINTY (Settles & Craven, 2008): This method is based on informativeness. Specifically, it queries the instance with most uncertainty. The uncertainty is measured by prediction confidence.

- ENTROPY (Lewis & Catlett, 1994): This method is based on informativeness. Specifically, it queries the instance with the highest entropy.

- CORESET (Sener & Savarese, 2018): This method is based on representativeness. Specifically, it queries the instance minimize the core-set loss.

- QUIRE (Huang et al., 2014): This method is a hybrid method, which queries the instance with informativeness and representativeness.

- WMOCUAL (Zhao et al., 2021): This method is a hybrid method, which queries the instance based on the weighted form of MOCU.

In our experiment, we use QUIRE and CORESET in (Tang et al., 2019). For each data set, we randomly choose 20% of instances for testing. We randomly choose 5 positive instances and 5 negative instances as initial labeled instances. In each iteration, we query one instance from $\mathbb{U}$ and add it to $\mathbb{L}$. For data sets with instances less than 1000, we query 100 instances in total. For data sets with instances less than 5000, we query 300 instances in total. For data sets with instances more than 5000, we query 500 instances in total.

## 4.2 PERFORMANCE

We initially concatenate AQOT with SVM rather than with all three classifiers to demonstrate the results. Figure 3 shows the performance of eight methods on six data sets in terms of accuracy. In phoneme, QUIRE needs more than 2 hours to get the result, so there is no result of QUIRE in the corresponding figure.

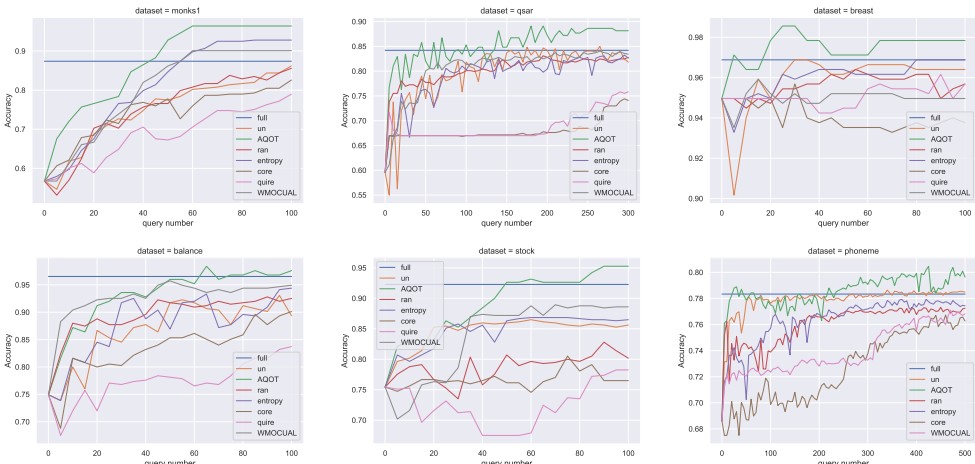

Figure 3: Results on six data sets in terms of accuracy. QUIRE spends too much time (over 2 hours), so its result is not shown in the corresponding figure.

It can be seen from the result that on all data sets AQOT outperforms most of the baselines and achieves the best performance in all cases.

## 4.3 PARAMETER ANALYSIS

As introduced above, $\lambda$ has influence on the solution of the OT problem. If $\lambda$ is too small, the classifier degenerates to the original OT. Conversely, if $\lambda$ is too large, the coefficient vector becomes almost uniformly distributed. From the experimental point of view, a reasonable range of $\lambda$ is between 0.1 and 1. We concatenate AQOT with three classifiers and consider values of $\lambda \in \{0.1, 0.5, 1\}$. Table 1 summarizes the performance of the nine methods on six data sets in terms of F1 score based on ten trials. The win/tie/loss counts are summarized in the last three rows.

It can be seen from the result that AQOT methods outperform most of the baselines regardless of the value of $\lambda$, and the performance is related to the classifier. Observing the results, we can also find that under the appropriate regularization parameter, AQOT is not sensitive to $\lambda$.

Another parameter $\alpha$ controls the weight of certainty and uncertainty. If $\alpha$ equals to 0, the term of certainty disappears. Conversely, if $\alpha$ equals to 1, the term of uncertainty disappears. We concatenate AQOT with three classifiers and consider values of $\alpha \in \{0.1, 0.2, 0.3, 0.4, 0.5, 0.6, 0.7, 0.8, 0.9, 1\}$.Figure 4 shows the performance of three AQOT methods on six data sets in terms of F1 score.

It can be seen from the result that a too small $\alpha$ or too big $\alpha$ will decrease the performance of AQOT method. A suitable range for $\alpha$ falls between 0.4 and 0.6 based on the data set, which means the importance of certainty and uncertainty is similar in most of the cases.

## 4.4 RUN TIME COMPARISON

We compare the running time to show AQOT is efficient. All algorithms are implemented in Python 3.7 on a personal computer with Intel i5-12500 2.5 GHz CPU and 16G RAM. Table 2 shows the result. It can be seen from the result that compared to two hybrid methods: QUIRE and WMOCU, our AQOT methods shows higher efficiency.

Table 1: Results on six data sets in terms of F1 score over 10 trials. The best result on each data set is indicated in bold. The win/tie/loss counts are summarized in the last three rows. (A wins B means A is significantly better than B based on a pair-wise t-test at a 0.05 significance level)

| Dataset | $\lambda$ | RAN | UN | EN | CORE | QUIRE | WMOCU | AQSVM | AQGBDT | AQNN |
|---|---|---|---|---|---|---|---|---|---|---|
| monks1 | 0.1 | .856±.010 | .868±.014 | .904±.012 | .868±.016 | .880±.025 | .902±.012 | .927±.014 | .973±.026 | **.982±.013** |
| | 0.5 | .856±.010 | .868±.014 | .904±.012 | .868±.016 | .880±.025 | .902±.012 | .936±.034 | **.980±.010** | .972±.015 |
| | 1 | .856±.010 | .868±.014 | .904±.012 | .868±.016 | .880±.025 | .902±.012 | .958±.034 | **.990±.010** | .975±.018 |
| qsar | 0.1 | .831±.033 | .818±.039 | .827±.031 | .815±.023 | .811±.028 | .852±.012 | .861±.020 | .874±.015 | **.895±.032** |
| | 0.5 | .831±.033 | .818±.039 | .827±.031 | .815±.023 | .811±.028 | .852±.012 | .885±.014 | **.897±.019** | .885±.037 |
| | 1 | .831±.033 | .818±.039 | .827±.031 | .815±.023 | .811±.028 | .852±.012 | .881±.025 | .883±.035 | **.894±.034** |
| balance | 0.1 | .945±.012 | .933±.030 | .933±.012 | .942±.013 | .923±.022 | .942±.006 | **.976±.014** | .955±.007 | .958±.008 |
| | 0.5 | .945±.012 | .933±.030 | .933±.012 | .942±.013 | .923±.022 | .942±.006 | **.980±.011** | .952±.012 | .957±.010 |
| | 1 | .945±.012 | .933±.030 | .933±.012 | .942±.013 | .923±.022 | .942±.006 | **.960±.028** | .956±.012 | .957±.013 |
| stock | 0.1 | .859±.020 | .856±.033 | .868±.019 | .874±.019 | .735±.024 | .877±.022 | .903±.034 | **.957±.028** | .948±.017 |
| | 0.5 | .859±.020 | .856±.033 | .868±.019 | .874±.019 | .735±.024 | .877±.022 | .917±.029 | .945±.031 | **.952±.024** |
| | 1 | .859±.020 | .856±.033 | .868±.019 | .874±.019 | .735±.024 | .877±.022 | .943±.022 | **.962±.022** | .945±.013 |
| breast | 0.1 | .961±.012 | .960±.011 | .969±.008 | .959±.016 | .962±.012 | .955±.020 | **.979±.012** | .978±.007 | .977±.012 |
| | 0.5 | .961±.012 | .960±.011 | .969±.008 | .959±.016 | .962±.012 | .955±.020 | .976±.011 | .975±.005 | **.979±.017** |
| | 1 | .961±.012 | .960±.011 | .969±.008 | .959±.016 | .962±.012 | .955±.020 | **.978±.010** | .978±.006 | .975±.011 |
| phoneme | 0.1 | .773±.014 | .788±.010 | .784±.037 | .765±.015 | N/A | .768±.013 | .803±.012 | **.863±.019** | .804±.014 |
| | 0.5 | .773±.014 | .788±.010 | .784±.037 | .765±.015 | N/A | .768±.013 | .820±.038 | **.858±.021** | .802±.011 |
| | 1 | .773±.014 | .788±.010 | .784±.037 | .765±.015 | N/A | .768±.013 | .815±.027 | **.863±.017** | .805±.012 |
| AQSVM: w/t/l | | 18/0/0 | 17/1/0 | 18/0/0 | 17/1/0 | 18/0/0 | 16/2/0 | | | |
| AQGBDT: w/t/l | | 15/3/0 | 18/0/0 | 18/0/0 | 17/1/0 | 18/0/0 | 18/0/0 | | | |
| AQNN: w/t/l | | 15/3/0 | 17/1/0 | 18/0/0 | 17/1/0 | 18/0/0 | 18/0/0 | | | |

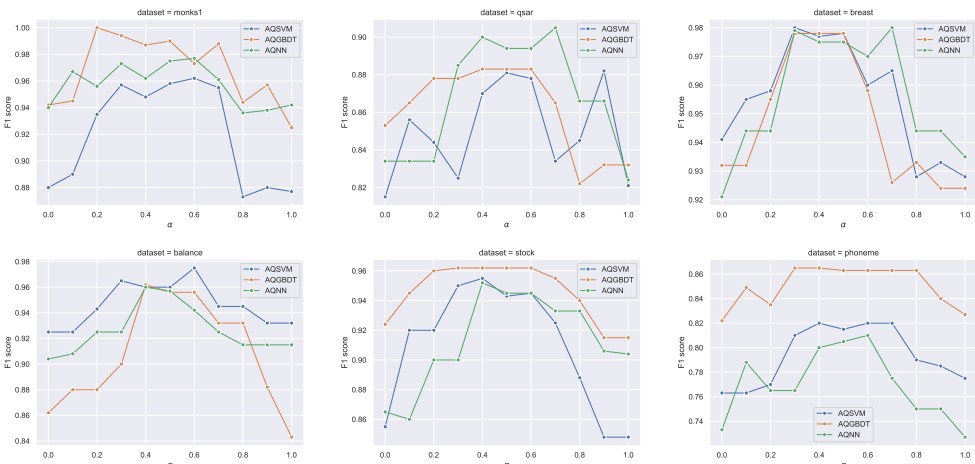

Figure 4: Parameter analysis on six data sets. We adjust the value of $\alpha$ to show the influence of $\alpha$.

### 4.5 ABLATION EXPERIMENTS

As introduced above, our dynamic query score consists of three terms: certainty, uncertainty and dynamic adjustment term. We compare our AQOT method with methods lacking each of these three terms respectively. Table 3 presents the results. From the table, we can see that the method with dynamic adjustment surpasses the performance of the method regardless of lacking which term.

## 5 CONCLUSION

In this paper, we proposed an efficient active query strategy AQOT. We establish two OT models from unlabeled instances to positive and negative instances respectively. We evaluate certainty of instances by the standard deviation of coefficient vector and evaluate uncertainty by the difference of two highest posterior probabilities. We query instances by weighing certainty and uncertainty

Table 2: Results on six data sets in terms of running time (in seconds).

| Data set | RAN | UN | EN | CORE | QUIRE | WMOCU | AQSVM | AQGBDT | AQNN |
|---|---|---|---|---|---|---|---|---|---|
| monks1 | 0.107 | .0.257 | 1.187 | 0.160 | 286 | 173 | 0.708 | 3.63 | 7.53 |
| qsar | 0.722 | 2.78 | 9.87 | 1.69 | 628 | 546 | 5.67 | 38.7 | 48.3 |
| balance | 0.106 | 0.302 | 1.62 | 0.171 | 45.1 | 175 | 0.745 | 4.352 | 11.9 |
| stock | 0.109 | 0.334 | 1.90 | 0.209 | 417 | 212 | 0.977 | 4.60 | 11.0 |
| breast | 0.102 | 0.208 | 1.41 | 0.164 | 69.3 | 59.8 | 0.755 | 3.47 | 5.97 |
| phoneme | 4.83 | 32.2 | 192 | 0.171 | N/A | 1250 | 88.3 | 127 | 138 |

Table 3: F1 score of AQOT method and methods without three terms respectively over 10 trials. Method-1 denotes by the method lacking the certainty term. Method-2 denotes by the method lacking the uncertainty term. Method-3 denotes by the method lacking the dynamic adjustment term. ● indicates the performance of AQOT is significantly better than the compared method (pairwise t-test at 0.05 significance level).

(a) results on SVM

| Data set | AQSVM | AQSVM-1 | AQSVM-2 | AQSVM-3 |
|---|---|---|---|---|
| monks1 | .958±.034 | .880±.021● | .877±.012● | .921 ±.021● |
| qsar | .881±.025 | .815±.021● | .821±.012● | .860±.011 |
| balance | .960±.028 | .925±.027● | .932±.014● | .945±.022 |
| stock | .943±.022 | .855±.012● | .848±.021● | .839±.014● |
| breast | .978±.010 | .941±.014● | .928±.007● | .960±.016 |
| phoneme | .815±.027 | .763±.021● | .775±.026● | .800±.015● |

(b) results on GBDT

| Data set | AQGBDT | AQGBDT-1 | AQGBDT-2 | AQGBDT-3 |
|---|---|---|---|---|
| monks1 | .990±.010 | .942±.017● | .925±.023● | .933±.008● |
| qsar | .883±.035 | .853±.010● | .832±.025● | .812±.021● |
| balance | .956±.012 | .862±.011● | .843±.053● | .887±.012● |
| stock | .962±.022 | .924±.035● | .915±.014● | .903±.021● |
| breast | .978±.006 | .932±.025● | .924±.012● | .922±.015● |
| phoneme | .863±.017 | .822±.029● | .827±.018● | .842±.015● |

(c) results on NN

| Data set | AQNN | AQNN-1 | AQNN-2 | AQNN-3 |
|---|---|---|---|---|
| monks1 | .975±.018 | .940±.015● | .942±.012● | .947±.023● |
| qsar | .894±.034 | .834±.022● | .824±.012● | .852±.017● |
| balance | .957±.013 | .904±.008● | .915±.012● | .914±.028● |
| stock | .945±.013 | .865±.016● | .904±.011● | .896±.008● |
| breast | .975±.011 | .921±.021● | .935±.022● | .955±.010● |
| phoneme | .805±.012 | .733±.017● | .727±.023● | .725±.019● |

with encouraging early exploration with taking instance distribution into account. Moreover, AQOT shows high efficiency compared to other hybrid methods. We concatenate it with multiple classifiers to show it is a broad-spectrum strategy.

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
