# OpenReview forum: "An Efficient Query Strategy for Active Learning via Optimal Transport"
_ICLR.cc/2024/Conference — Submitted to ICLR 2024_

### Official Review · Reviewer_2t83 · 2023-10-30

**Soundness:** 2 fair
**Presentation:** 3 good
**Contribution:** 2 fair
**Rating:** 5
**Confidence:** 3

**Summary:**

This paper studies active learning. Existing methods often focus on either informativeness or representativeness, which can lead to limitations. To address this, the authors propose an efficient strategy called  AQOT, based on the concept of Optimal Transport (OT). AQOT leverages OT to measure the difference between instance distributions, allowing for a balance between informativeness and representativeness. The method efficiently selects instances based on the sparsity of OT solutions and can be adapted to multiple classification models. Experimental results indicate that AQOT outperforms other active learning approaches in terms of both effectiveness and efficiency.

**Strengths:**

The idea of leveraging OT for both positive and negative instances is interesting and the design query strategy by examining the differences in the distributions of coefficient vectors between these two models. Authors also conducted many experiments to verify their methods.

**Weaknesses:**

However, the technical novelity of this paper is limited. The idea of using OT in active learning is not very fresh and query strategy looks a little bit of engineering. In the experiment section, the used baselines looks like out of date and would suggest add some baseline of year 2022.

**Questions:**

I am wondering how is your method associated with representativeness?

---

### Official Review · Reviewer_A9xS · 2023-10-31

**Soundness:** 2 fair
**Presentation:** 2 fair
**Contribution:** 2 fair
**Rating:** 5
**Confidence:** 3

**Summary:**

This paper introduces an active query strategy named AQOT that effectively combines informativeness and representativeness for active learning. It utilizes Optimal Transport (OT) models to measure the differences between distributions of unlabeled instances, allowing for informed query selection. The approach considers both certainty and uncertainty, encouraging early exploration while maintaining high efficiency.

**Strengths:**

1. The use of Optimal Transport to combine informativeness and representativeness is a novel and innovative approach to active learning.

2. The dynamic adjustment term encourages early exploration, which can lead to quicker adjustments of decision boundaries.

**Weaknesses:**

1. The proposed idea is quite similar to several previous works on domain adaptation by using OT (see ``Nicolas Courty, Rémi Flamary, Devis Tuia, Alain Rakotomamonjy: Optimal Transport for Domain Adaptation'' for example). Though DA is a different problem with active learning,  both of them are used for solving the limited label issue. So the idea proposed in this paper seems not that novel.

2. The experiments are not sufficient. Several new active learning algorithms are not tested, like Jordan T. Ash, Chicheng Zhang, Akshay Krishnamurthy, John Langford, Alekh Agarwal: Deep Batch Active Learning by Diverse, Uncertain Gradient Lower Bounds. ICLR 2020

3. What are positive and negative instances? You mean you only consider binary classification?

4. Lack of Theoretical Analysis: While the paper presents compelling experimental results, it lacks an in-depth theoretical analysis of the AQOT approach. A more comprehensive theoretical foundation would enhance the paper's contributions.

5. The use of Optimal Transport models may introduce computational complexity.

**Questions:**

1. Considering that the sparsity of the coefficient vector is introduced by the parameter $\lambda$ in the Sinkhorn algorithm, intuitively, a sparser coefficient vector implies the presence of nearby samples in the labeled dataset. Have you ever considered the correlation between the sparsity of the coefficient vector and the nearest neighbors of the instances?

2. Are there any plans to further optimize the AQOT method to address the computational complexity associated with the use of the Optimal Transport model?

3. Are there any plans to further explore the theoretical foundations of the AQOT method?

---

### Official Review · Reviewer_xjm3 · 2023-10-31

**Soundness:** 2 fair
**Presentation:** 2 fair
**Contribution:** 1 poor
**Rating:** 3
**Confidence:** 5

**Summary:**

The paper proposes an active learning query strategy based on optimal transport. Unlike traditional methods that focus on informativeness or representativeness, the proposed strategy directly aims at optimal transport. The proposed strategy utilizes existing methods to solve the optimal transport problem of the unlabeled set to the positive set and the negative set respectively, and proposes a dynamic query score based on the coefficient matrices.

**Strengths:**

The paper has a good motivation of studying active learning from the optimal transport perspective. The experimental results show a decent performance on several datasets compared to a few baselines and the improvement is statistically verified.

**Weaknesses:**

Although the paper might aim at optimal transport more directly, this is not the first attempt at utilizing the idea of optimal transport in active learning ([1] Shui, Changjian, et al. "Deep active learning: Unified and principled method for query and training." International Conference on Artificial Intelligence and Statistics. PMLR, 2020.). Compared to existing methods, the proposed method seems limiting and the evaluation is also not very convincing. The dynamic score is still a heuristic method combining uncertainty and diversity, which is quite similar to lots of existing works ([2] Huang, Sheng-Jun, Rong Jin, and Zhi-Hua Zhou. "Active learning by querying informative and representative examples." Advances in neural information processing systems 23 (2010).). The main results focus on binary classification problems and there does not seem to be a way to generalize. The compared baselines are mostly older methods, with more recent deep active learning (WAAL from [1], classic methods like BatchBALD [3] Kirsch, Andreas, Joost Van Amersfoort, and Yarin Gal. "Batchbald: Efficient and diverse batch acquisition for deep bayesian active learning." Advances in neural information processing systems 32 (2019). and so on) or even binary classification active learning ([4] Raj, Anant, and Francis Bach. "Convergence of uncertainty sampling for active learning." International Conference on Machine Learning. PMLR, 2022., etc.) baselines missing. The overall contribution is not significant because of the above reasons.

**Questions:**

Please see weaknesses.

---

### Official Review · Reviewer_nES8 · 2023-11-05

**Soundness:** 3 good
**Presentation:** 3 good
**Contribution:** 2 fair
**Rating:** 5
**Confidence:** 4

**Summary:**

This paper proposed a new active learning query score for pool-based setting. The score is composed by three parts: certainty, uncertainty, and dynamic adjustment term. The main novelty comes from the certainty term, which is measured by the absolute difference of standard deviations of coefficient vectors of the optimal transports from the candidate unlabeled point to positive set and negative set. The intuition is that the coefficient vector is sparse and hence is of high variance when transporting to the set with the same label, but more uniform if it’s transporting to a different class. The optimal transport can be efficiently solved by an existing algorithm after adding entropy regularization.  Uncertainty is simply the inverse of margin, and the dynamic adjustment is the sum of distance to a labeled set, weighted by a parameter that diminishes as iterations go, which encourages exploration in the beginning. Experiments show superior performance on six UCI datasets with 3 typical learning algorithms such as SVM, GBDT, NN. Ablation studies are conducted to demonstrate the importance of each term and how sensitive the entropy regularization parameter is and the empirical optimal range of the relative importance between the certainty and uncertainty terms.

**Strengths:**

The idea of leveraging optimal transport in the proposed manner for active learning seems novel (but may survey literature on optimal transport for active learning). The paper is well written and easy to follow. The experiments are fairly comprehensive with ablation studies.

**Weaknesses:**

The proposed score is a heuristic combination of several scores measuring various factors with tunable weights — I don’t think this is a principled approach.
From the abstract seems the main motivation is to have a score measuring both representativeness and informativeness, but it’s not clearly written which part measures representativeness, I would think it’s the “certainty” term since it seems to avoid outliers, but from paper it’s for informativeness, but the “uncertainty” term is also for informativeness. It would be great to have more studies on how the “certainty” or “confidence” term is related to the “uncertainty” term, perhaps with a visual demonstration of what kinds of points the sum of the two scores are favoring.

**Questions:**

In experiments, baseline “FULL” is using the entire labeled set for training? And the proposed AL algorithm querying much less labels could often surpass “FULL” from Figure 3? If so, seems interesting to understand the reasons behind. E.g., Is it because AQOT is better at filtering out noise?
If I understand the intuitive correctly, one simple alternative to the “certainty” score I can think of is: absolute diff of average distance to positive and negative labeled set, wonder if this could achieve the same effect?

---

### Meta-Review · Area_Chair_ZXKq · 2023-12-07

**Metareview:**

The authors consider the use of ideas from optimal transport in an active learning setting. To this end, the authors propose an active learning strategy called active query by optimal transport, discuss details regarding its implementation, and evaluate it empirically.

The reviewers agree that this work is of interest to the ICLR community and that the general idea of incorporating optimal transport into active learning is sound and worth exploring.

However, the reviewers also noted several perceived weaknesses with the manuscript as submitted:

- lack of clarity regarding connections to existing work, with some reviewers questioning the novelty and significance of the proposals here
- a perceived lack of sufficient empirical evaluation / concerns regarding the design of the empirical study

**Justification For Why Not Higher Score:**

The reviewers identified several serious weaknesses with the submitted manuscript, which were not sufficiently addressed during the author response period (the authors did not elect to respond).

**Justification For Why Not Lower Score:**

N/A

---

### Decision · Program_Chairs · 2024-01-16

Reject